# Amazon River plume habitats shape planktonic cnidarian assemblages in the Western Atlantic

Everton Giachini Tosetto [1,2,3]*, Sigrid Neumann-Leitão [3], Moacyr Araujo [3,4], Djoirka Minto Dimoune [3], Arnaud Bertrand [1,2,3,5]*, Miodeli Nogueira Júnior [6]*

**1** Institut de Recherche pour le Développement, Sète, France, **2** MARBEC, Univ Montpellier, CNRS, IFREMER, IRD, Sète, France, **3** Departamento de Oceanografia, Universidade Federal de Pernambuco, Recife, Brazil, **4** Brazilian Research Network on Global Climate Change (Rede CLIMA), São José dos Campos, Brazil, **5** Departamento de Pesca e Aquicultura, Universidade Federal Rural de Pernambuco, Recife, Brazil, **6** Departamento de Sistemática e Ecologia, Universidade Federal da Paraíba, João Pessoa, Brazil

* evertontosetto@hotmail.com (EGT); arnaud.bertrand@ird.fr (AB); miodeli@gmail.com (MNJ)

**Data Availability Statement:** The data underlying the results presented in the study are available in SEANOAE (https://doi.org/10.17882/93375).

## Abstract

The impact of the Amazon River freshwater plume on planktonic cnidarians over neritic and oceanic provinces is unknown. To provide further knowledge we took advantage of an oceanographic cruise performed in October 2012 in the Western Atlantic off the North Brazilian coast (8˚N, 51˚W—3.5˚S, 37˚W). A complex and dynamic system was observed, with strong currents and eddies dispersing the plume over a large area. Our results show that the Amazon River shapes marine habitats with a thin highly productive surface layer compressed by a deeper oxygen minimum zone both over the shelf and in the open ocean. We hypothesized that such habitat structure is particularly advantageous to planktonic cnidarians, which have low metabolic rates, being able to survive in hypoxic zones, resulting in high species richness and abundance. Over the shelf, distinctions were sharp and the area under the influence of the plume presented a diverse assemblage occurring in large abundance, while outside the plume, the hydromedusa *Liriope tetraphylla* was dominant and occurred almost alone. Divergences in the oceanic province were less pronounced, but still expressive being mostly related to the abundance of dominant species. We concluded that Amazon River plume is a paramount physical feature that profoundly affects the dynamics of the mesoscale habitat structure in the Western Equatorial Atlantic Ocean and that such habitat structure is responsible for shaping planktonic cnidarian assemblages both in neritic and oceanic provinces.

## Introduction

In marine environments, planktonic cnidarians respond to changes in physical and biogeo-chemical environments over a wide range of spatiotemporal scales. Global cnidarian diversity, distribution and abundance patterns are closely related to oceanographic dynamics and water masses, and to climate patterns [1–3]. Currents, eddies, fronts of water masses, upwelling and

**Funding:** The author received no specific funding for this work.

**Competing interests:** The authors have declared that no competing interests exist.

other physical processes can drive their distribution at regional scales [1,4–9]. Finally, responses to local changes in the environment, such as prey availability and water temperature and salinity, may determine species local occurrence and abundance [10–12].

In the past, planktonic cnidarians were often set aside in traditional zooplankton studies. However, their high feeding rates and significant role as predators in the trophic web, associated with large population blooms, which occur in the life cycles of many species, have the potential to control the pelagic community and collapse fisheries [13,14]. Therefore, interest in pelagic cnidarian ecology, their responses to environmental conditions, such as bloom trigger mechanisms, and their role in ecosystem functioning as predators and prey for higher tropic levels increased in recent decades [14–17]. However, knowledge on the ecology of planktonic cnidarians is still scarce in low latitude systems such as the Western Equatorial Atlantic Ocean.

As a western boundary current system, a strong current, the North Brazil Current (NBC) flows coastward in this region [18,19]. Such feature results in intrusions of oligotrophic oceanic waters and associated planktonic cnidarian fauna over the continental shelf in other western boundary systems [20,21]. Despite the oligotrophic scenario brought by the NBC, the Amazon River discharges up to $2.4 \times 10^5 \ m^3 s^{-1}$ of freshwater (approximately 20% of global freshwater run-off) with organic matter, nutrients and sediments over the area [22,23]. The Amazon discharge creates a surface plume (Amazon River plume, hereafter ARP) of low-salinity, high nutrients, and suspended and dissolved materials that reaches thousands of kilometers in the Equatorial and North Atlantic and Caribbean Sea [24–26] with strong influence on ecological processes. The large discharge of nutrients enhances primary production and phytoplankton concentration and possibly the whole marine community by a bottom-up effect, as observed with planktonic decapods [27,28]. The low salinity brackish environment may affect the spatial distribution of marine animals such as planktonic cnidarians, which usually are associated with specific environmental saline conditions and water masses [5,9,29,30]. Additionally, bellow the plume, low oxygen levels are observed due the high rates of sinking organic matter mineralization [31,32].

Freshwater run-off from small rivers are known to affect the structure of planktonic cnidarian assemblage in coastal areas [33–39] where species have distinct responses to the salinity gradient. In the Western Equatorial Atlantic, the ARP extends far offshore and its influence on planktonic cnidarian distribution and abundance may occur thousands of kilometers away from the river mouth. However, effects of such major river plumes on cnidarians were never specifically evaluated. In this study, we investigate the hypothesis according to which unlike other western boundary current systems, the ecological processes induced by the spread of the ARP, drive the structure of planktonic cnidarian assemblages both over the shelf and offshore. For that purpose, we used a comprehensive set of data collected from neritic and oceanic provinces in the Western Equatorial Atlantic, both inside and outside the influence of the ARP, in order to understand the response of the planktonic cnidarian assemblage to this unique physical and biogeochemical environment.

## Materials and methods

### Study area

The study area was along the North Brazilian continental shelf between the Amazon and Oyapok river mouths and in Equatorial Atlantic oceanic waters, ranging between 8˚N, 51˚W and 3.5˚S, 37˚W (Fig 1). In the area, the continental shelf reaches up to 300 km wide and the shelf break occurs around 120 m depth [40,41]. The large freshwater discharge of the Amazon River creates an extensive surface plume. Strong ocean currents, eddies, wind fields and high tidal variation in the North Brazilian continental shelf and adjacent oceanic waters shape the

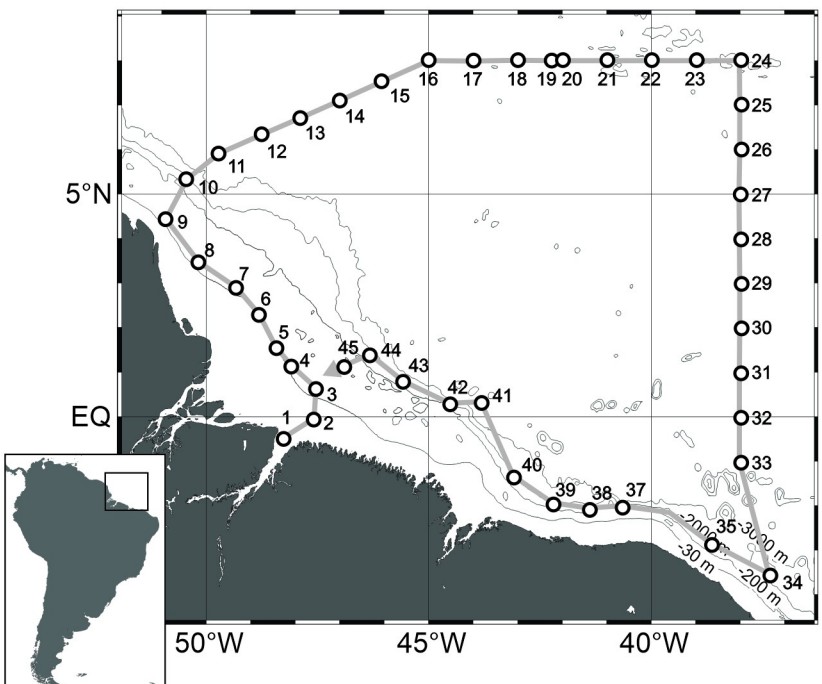

**Fig 1. Geographic location of the study area in the North Brazilian continental shelf and adjacent Western Equatorial Atlantic Ocean, showing the sampled stations.**

dispersion of the ARP, resulting in a highly dynamic system with high spatial and temporal variability [41–44]. Three general patterns occur in the spread of the ARP throughout the year. From January to April, the ARP flows continuously northwest along the Brazilian coast carried mainly by NE winds. From April to July, the ARP reaches the Caribbean region due to the higher discharge, NBC transport and SE winds. Finally, from August to December, when wind fields are weaker, the retroflection (i.e. a change in the flow direction) of the NBC, around 5 and 10˚N, disperses the ARP to the east, feeding the North Equatorial Counter Current [NECC; 43,45]. During this period the plume that can exceed $10^6$ km$^2$, reaching longitudes to the east as far as 25˚W [46].

## Sampling

Data were obtained during the oceanographic cruise *Camadas Finas* III, aboard the research vessel NHo. Cruzeiro do Sul—H38 (DHN/Brazilian Navy). It was performed in 9–31 October 2012, the period when most of the ARP is retroflexed and transported eastward by the NECC [43,45]. Zooplankton was sampled at 44 stations in oblique hauls, using Bongo nets with 120 and 300 μm mesh and 0.3 and 0.6 m mouth diameter, respectively. Tows were performed at approximately 2 knots from near bottom to surface on the continental shelf, and from 200 m to the surface in the open ocean. Nets were fitted with a flowmeter (Hydro-Bios) to estimate the filtered volume. Samples were fixed with 4% formaldehyde buffered with sodium tetraborate (0.5 gL$^{-1}$). Current speed and directions were recorded along all the track of the ship by a Teledyne RD Ocean Surveyor ADCP. Data from the first 100 m of the water column were integrated each 30 km along the track. Salinity, temperature (˚C), density ($\sigma_t$), dissolved oxygen (mgL$^{-1}$) and fluorescence vertical profiles were obtained at stations where zooplankton was sampled with a Seabird SBE 25 Sealogger CTD profiler [47].

In laboratory, whole zooplankton samples were analyzed under stereomicroscope and specimens were identified [mainly following 48,49] and counted. A general taxonomic overview of planktonic cnidarians of the area was previously reported [50]. Abundances were standardized in number of individuals per 100 m$^{-3}$ for medusae and number of colonies per 100 m$^{-3}$ for siphonophores. For calycophorans, the number of anterior nectophores was used for estimating the polygastric stage abundance, and eudoxid bracts for the eudoxid stage abundance [e.g. 30,51]. For physonects and the calycophoran *Hippopodius hippopus*, number of colonies were roughly estimated by dividing the number of nectophores by 10 [52]. Wet weight from 120 μm mesh samples (including cnidarians) measured by gravimetry after removing the excess water with blotted paper [53] were used as an indirect estimator of zooplankton biomass (mostly copepods, cladocerans and other crustaceans).

## Data analysis

The ARP was delimited by the isohaline of 35 and density <22 ($\sigma_T$). Tropical Surface Water (TSW) and South Atlantic Central Water (SACW) masses were delimited by the isobar of 24.5 ($\sigma_T$) [43,54,55]. Cyclonic and anticyclonic eddies were respectively identified by negative and positive sea levels anomalies in daily L4 satellite data, measured by multi-satellite altimetry observations over Global Ocean, produced by SSALTO/DUACS and distributed by European Union Copernicus Marine Service Information. Dominant current in each station was determined by the overall direction observed in the ADCP data and classified as NBC, retroflection area (RETRO) and NECC.

The abundance, species richness and structure of the planktonic cnidarian assemblage was statistically similar in samples from nets with 120 and 300 μm meshes [56]. Thus, we merged the data from both meshes for statistical analysis. Station 1, the was the only spot sampled inside the river estuary and was excluded from the data analysis. For statistical tests, abundance data was transformed by log (x+1). Analyses of Variance (ANOVA) were performed to test for differences in abundance of the most abundant planktonic cnidarian species, according to the province (neritic and oceanic) and influence of the ARP. Tukey post-hoc test was used to identify the domains that differed when ANOVA was significant. Spatial patterns in planktonic cnidarian assemblage abundance were identified by a hierarchical cluster analysis (Bray-Curtis similarity matrix). A Similarity Percentage (SIMPER) analysis was performed in order to identify key species and their contribution to similarity within the groups defined in the cluster analysis.

To identify associations between representative planktonic cnidarian taxa (species occurring in more than 21 stations and species with high abundance in few stations) and the physical environment, a constrained ordination analysis was performed. Detrended Canonical Correspondence Analysis (DCCA) revealed a small length of environmental gradients (<3), indicating that a linear method was appropriate, and thus Redundancy Analysis (RDA) was selected [57]. Mesoscale physical processes were included as dummy categorical explanatory variables (neritic and oceanic habitats, presence of ARP, predominant current, presence of cyclonic and anticyclonic eddies). Zooplankton biomass from nets with 120 μm (considered as food availability for planktonic cnidarians), maximum value of fluorescence (as a proxy of biological productivity), maximum value of dissolved oxygen and surface temperature and salinity were included as continuous explanatory variables. Monte Carlo test was used to test the significance of the first and all canonical axes together [57].

Environmental and biological distribution maps were produced in Qgis 3.2.1. Cluster and SIMPER analysis were performed in Primer v.6 + PERMANOVA. DCCA and RDA were performed in CANOCO 4.5.

## Results

### Environmental background

The southern part of the studied area (reaching 3˚N) was dominated by the North Brazilian Current (NBC), flowing west and coastwards. Around 5˚N (station 10), the NBC was retroflexed northwards, where faster currents occurred. In the north of the study area, from 7.5˚N, 46˚W (station 15) to 4˚N, 38˚W (station 28) the North Equatorial Counter Current (NECC), flowing east, predominated (Fig 2a). One cyclonic eddy, causing surface divergence, occurred near the mouth of the Amazon River and stations 4, 5 and 6 were sampled under its influence. Stations 8, 13, 14, 17, 18 were sampled under the influence of three anticyclonic eddies and surface convergence (Fig 2b). Although another cyclonic eddy occurred in the west side of the study area, it was already dissipated in the day those stations were sampled.

Over the continental shelf, surface waters of stations 8 and 9 were influenced by the ARP resulting in salinities below 35 in the first 8 m of the water column (Figs 2c and 3a). In the oceanic province, stations 15 to 24 were influenced by the ARP, where the 35 isohaline ranged from 14 m depth at station 21 to 59 m depth at station 19. Outside the ARP, surface salinity was around 36 in all stations (Fig 2c).

Coastal surface waters were slightly warmer than adjacent oceanic areas, reaching 28.5˚C. In oceanic waters, surface temperature oscillated mainly with latitude with higher temperatures in north of the study area, reaching 29.8˚C (Fig 2d). Colder waters (<18˚C) occurred commonly around 120 m depth, below a strong thermocline, however intrusions in the upper layers were observed at stations 21 to 24 were it reached up to 60 m depth. Distinctly, close to the continental slope, warm surface water transposed down to 150 m depth (Fig 3b). These upwelling and downwelling features reflect the complex circulation system in the area. Following the temperature and salinity gradients, three water masses were observed in the first 200 m, ARP waters ($\sigma_T < 22$), Tropical Surface Waters (TSW) ($\sigma_T$ between 22 and 24.5) and South Atlantic Central Waters (SACW) ($\sigma_T > 24.5$; Figs 2e and 3c).

Fluorescence was higher in the surface layer of neritic stations influenced by the ARP, reaching 2.7 (Fig 2g). A small increment in fluorescence was also observed near the bottom at station 5, which was under the influence of a cold core cyclonic eddy. In the open ocean, a deep fluorescence maximum layer was observed in the boundary between TSW and SACW (Fig 3d). High zooplankton biomass occurred in the neritic area under influence of the ARP and in the region of the cold core cyclonic eddy, no clear spatial patterns were observed in other stations (Fig 2h). Dissolved oxygen was steady in the surface layer over the entire study area. In deeper waters, both in neritic and oceanic zones under the influence of the ARP, an oxygen minimum layer (<3 mg.l$^{-1}$) was observed between 50 and 200 m depth (Figs 2f and 3e).

### Species composition

A total of 91 taxa of planktonic cnidarians were collected (Table 1), corresponding to 2 scyphomedusae, 41 hydromedusae and 48 siphonophores. Furthermore, many unidentified cerinula, ephyrae and athorybia larval forms were collected. *Liriope tetraphylla* was the most frequent medusa, being present in 88.5% of the samples all over the study area, followed by *Aglaura hemistoma* (78.2%) and *Sminthea eurygaster* (41.4%). Among siphonophores, the most frequent were *Diphyes bojani* (88.5%), *Bassia bassensis* (80.5%), *Chelophyes appendiculata* (78.2%), *Abylopsis tetrago*na (75.9%), *Nanomia bijuga* (74.7%) and *Eudoxoides mitra* (75.6%; Table 1).

*Liriope tetraphylla* also dominated in abundance, representing 46% of all medusae. Other representative medusae were *Persa incolorata* (28.8%) and *A. hemistoma* (15.6%). Among

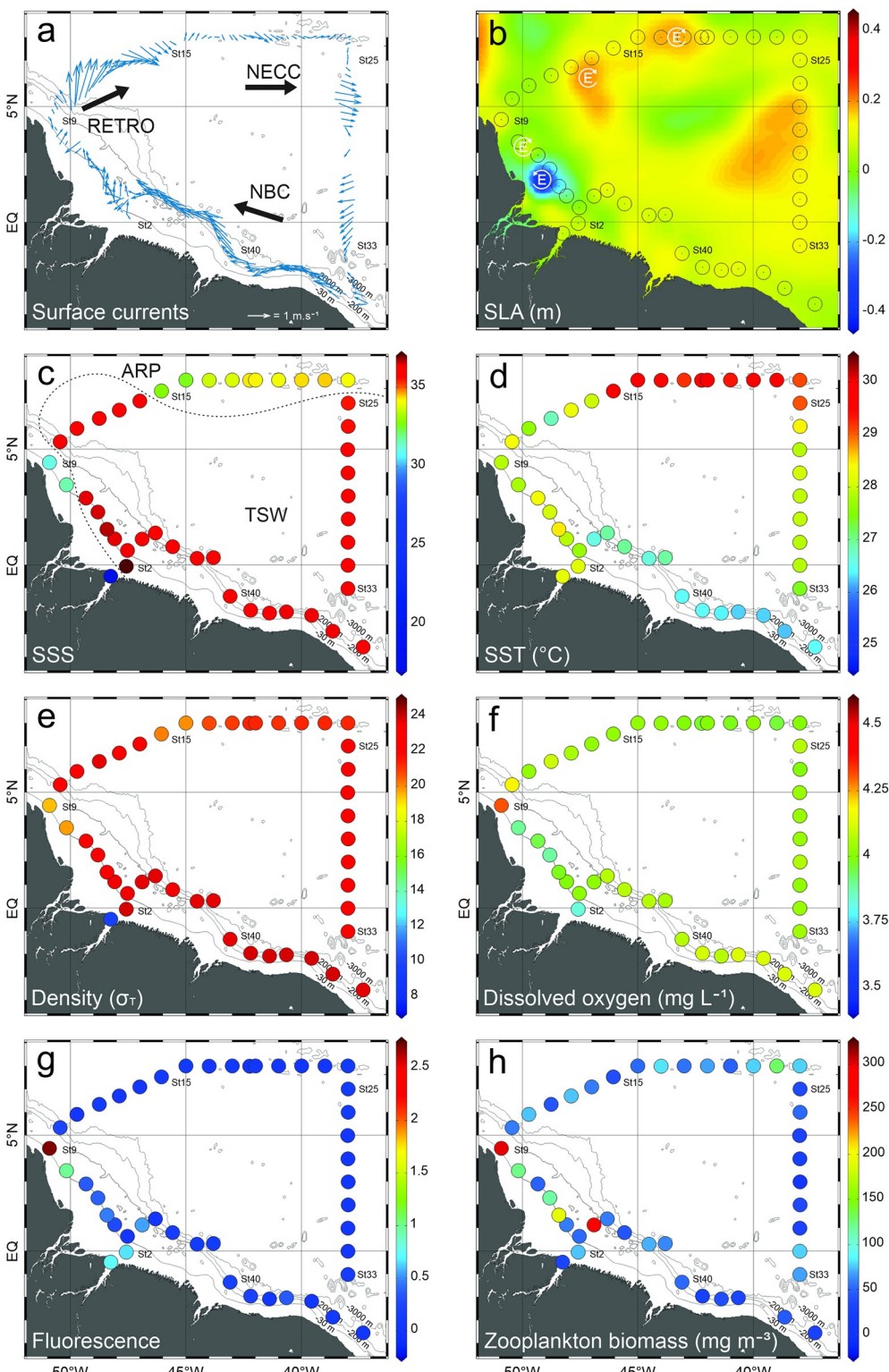

**Fig 2. (a)** Surface currents vectors and indicators of the predominant current in the area (NBC = North Brazilian Current; RETRO = North Brazilian Current retroflection; NECC = North Equatorial Counter Current). **(b)** Sea level anomalies and indicators of cyclonic and anticyclonic eddies. **(c)** Sea surface salinity and estimated position of 35 isohaline delimitating the Amazon River plume (ARP) and tropical surface water (TSW). **(d)** Sea surface temperature. **(e)** Surface density **(f)** Surface dissolved oxygen. **(g)** Surface fluorescence. **(h)** Total zooplankton biomass found in the

first 200 m of the water column (mean of samples collected with 120 and 300 μm mesh nets). All collected data are from October 2012.

siphonophores, *Enneagonum hyalinum* was the most abundant (19.1%), followed by *D. bojani* (18.2%), *C. appendiculata* (12.5%), *E. mitra* (10.8%) and *Muggiaea kochii* (10.6%).

## Spatial distribution patterns

Total medusa abundance was higher and more variable over the continental shelf, ranging from 1.4 to 1710 ind. 100 m$^{-3}$ (891.7±1161.3 in average). In this province, while medusa species richness was higher in the stations influenced by the ARP, high abundances occurred in stations both inside and outside of the ARP. In oceanic waters, highest medusa abundance occurred at stations located in the area influenced by the ARP and species richness was similar both inside and outside the influence of ARP (Fig 4).

The hydromedusa *L. tetraphylla* was widespread all over the sampled area. However, the species dominated with significant higher abundances (Fig 5; Table 2) the neritic stations outside the ARP. Its presence was also constant, although less abundant, in the retroflection area and in the NECC. Lower catches occurred in most oceanic stations under the influence of the NBC. *P. incolorata*, *Helgicirrha angelicae* and *Cunina octonaria* occurred in large abundances and almost exclusively in neritic stations under influence of the ARP. *Eutima marajoara* occurred in high abundance and exclusively at station 1, inside the estuary. *A. hemistoma* was the dominant medusa through most of the oceanic habitat. *S. eurygaster* was also representative in the open ocean, usually in lower densities. *Cytaeis* sp. 1 and *Solmundella bitentaculata* occurred scattered in low abundances all over the area (Fig 5).

Siphonophores species richness and overall abundance were more constant and typically high in oceanic stations, where it averaged 19.1±3.8 species and 193.8±107.7 ind. 100 m$^{-3}$, respectively. Differently, over the continental shelf, siphonophore species richness was considerably lower, typically <5 (3.6±3.4 in average), and they were absent at stations 2 and 4. Abundances in neritic stations also tended to be lower, but very high abundance (up to 3381.3 ind. 100 m$^{-3}$) occurred at station 9 (Fig 4).

*Enneagonum hyalinum* and *M. kochii* occurred in high abundance and almost exclusively in neritic stations under the influence of the ARP. Although widespread and abundant all over the area, *D. bojani* abundance was significantly higher in the oceanic stations under the influence of the ARP (Fig 6; Table 2). Also widespread, but with lower densities, *Diphyes dispar and N. bijuga* were more abundant in stations under the influence of the ARP or near its boundaries both in the neritic (significant for both species) and oceanic habitats (significant for *N. bijuga*, Fig 6; Table 2). All other abundant siphonophores occurred exclusively at oceanic stations and the neritic station 45, located near the shelf break (Fig 6). No clear spatial patterns were observed in the proportion of eudoxid and polygastric stages of calycophoran siphonophores.

## Assemblage structure

The cluster analysis depicted three main groups with low resemblance to each other (Fig 7). Group A, with 39.7% similarity, included the neritic stations without influence of the ARP, with the exception of the station 45. The group was represented mainly by *L. tetraphylla* in high abundance (Table 3). The two neritic stations under influence of the ARP belonged to Group B. With an average similarity of 69.5%, this group was mainly represented *P. incolorata*, *E. hyalinum*, *M. kochii* and *D. dispar* (Table 3).

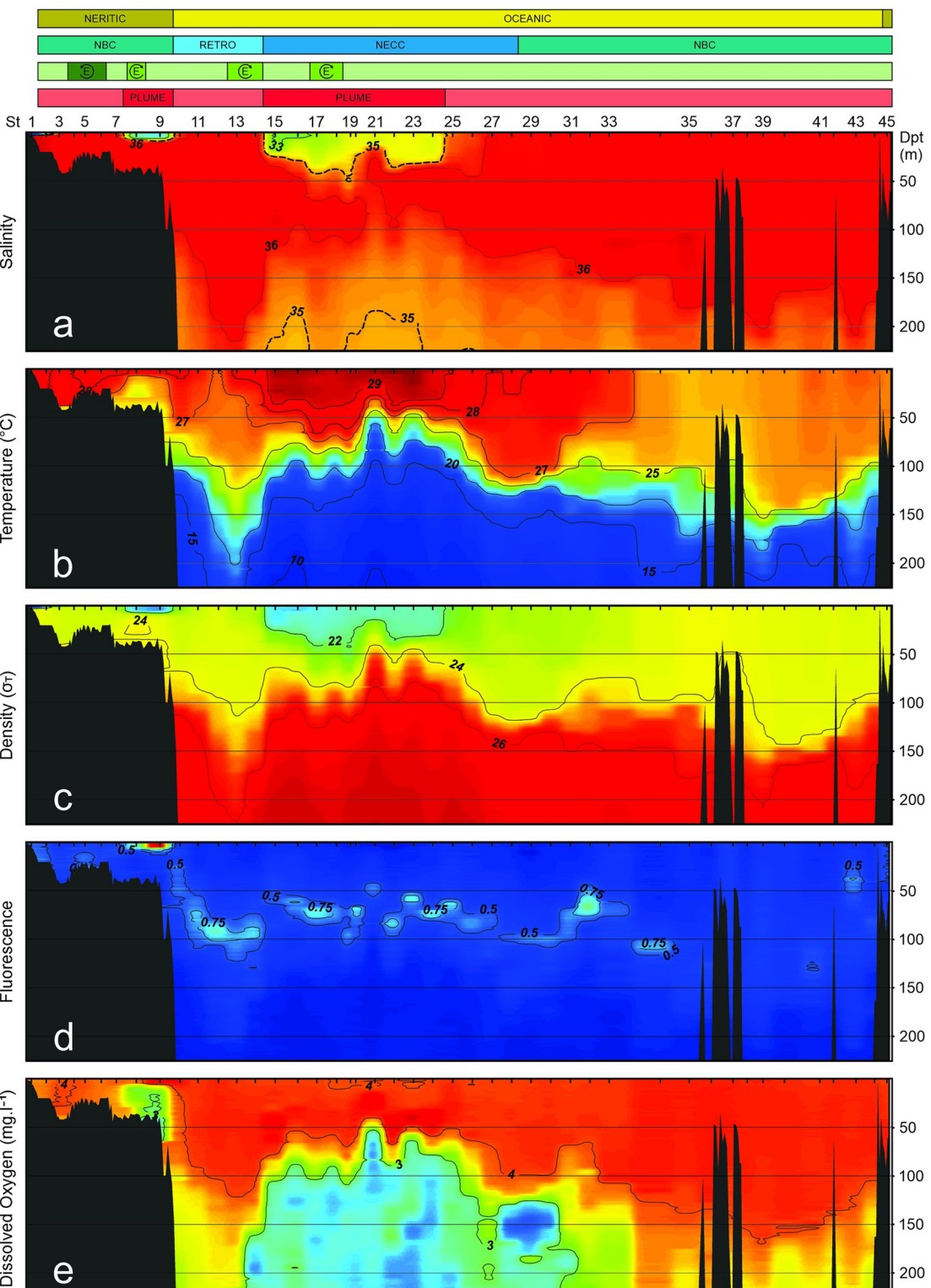

**Fig 3.** Vertical profiles and contours of **(a)** Salinity, traced line is the position of 35 isohaline delimitating the Amazon River plume. **(b)** Temperature. **(c)** Density. **(d)** Fluorescence. **(e)** Dissolved oxygen. Upper bars indicate the main mesoscale processes observed in the area. All collected data are from October 2012.

**Table 1. Basic statistics of planktonic cnidarian species from neritic and oceanic provinces in the Western Equatorial Atlantic Ocean off North Brazil.** Mean abundance (ind. 100 m$^{-3}$) per station and standard deviation, range of abundance and frequency of occurrence (f; considering both provinces; n = 87 samples). Species are sorted by frequency of occurrence. In all cases, both meshes were pooled.

| Species | Neritic | | Oceanic | | f |
|---|---|---|---|---|---|
| | Mean ± SD | Range of non-zero abundances | Mean ± SD | Range of non-zero abundances | |
| **Siphonophores** | | | | | |
| *Diphyes bojani* (Eschscholtz, 1825) | 19.7 ± 61.4 | 0.6–263.3 | 50.6 ± 81.4 | 2.8–471.4 | 88.5 |
| *Bassia bassensis* (Quoy & Gaimard, 1833) | 3.1 ± 8.33 | 1.6–30.5 | 19.7 ± 14.3 | 1.8–73.09 | 80.5 |
| *Chelophyes appendiculata* (Eschscholtz, 1829) | 8.6 ± 33.4 | 22.9–148.8 | 37.7 ± 35.6 | 3–161.9 | 78.1 |
| *Abylopsis tetragona* (Otto, 1823) | 2.1 ± 5.98 | 0.8–22.9 | 8.6 ± 8.3 | 0.9–40.5 | 75.9 |
| *Nanomia bijuga* (Delle Chiaje, 1844) | 3.3 ± 6.85 | 0.9–28.7 | 2.5 ± 2.9 | 0.1–13.5 | 74.7 |
| *Eudoxoides mitra* (Huxley, 1859) | 1 ± 2.94 | 0.8–11.4 | 34.1 ± 40.1 | 0.3–214.6 | 73.6 |
| *Abylopsis eschscholtzii* (Huxley, 1859) | 1.9 ± 7.67 | 2.9–34.3 | 7 ± 12.3 | 0.3–92.7 | 69 |
| *Sulculeolaria chuni* (Lens & van Riemsdijk, 1908) | 7.4 ± 28.35 | 22.9–125.9 | 7.6 ± 8.3 | 0.2–33.4 | 69 |
| *Eudoxoides spiralis* (Bigelow, 1911) | 1.7 ± 7.68 | 34.3–34.3 | 13.3 ± 21.3 | 0.7–123.1 | 60.9 |
| *Diphyes dispar* Chamisso & Eysenhardt, 1821 | 72.1 ± 187.61 | 0.7–768.7 | 1.7 ± 2.9 | 0.3–15.2 | 57.5 |
| *Agalma okenii* Eschscholtz, 1825 | 0.4 ± 1.71 | 7.6–7.6 | 1.1 ± 1.4 | 0.1–6.8 | 54 |
| *Sulculeolaria turgida* (Gegenbaur, 1854) | - | - | 0.9 ± 1.5 | 0.1–7 | 37.9 |
| *Ceratocymba leuckartii* (Huxley, 1859) | - | - | 1 ± 1.5 | 0.1–5.9 | 36.8 |
| *Lensia campanella* (Moser, 1917) | 0.4 ± 1.6 | 1.56–7.2 | 1 ± 1.7 | 0.3–7.6 | 34.5 |
| *Agalma elegans* (Sars, 1846) | - | - | 0.6 ± 1.2 | 0.3–6.7 | 29.9 |
| *Sulculeolaria biloba* (Sars, 1846) | - | - | 0.8 ± 1.4 | 0.4–5.4 | 29.9 |
| *Cordagalma ordinatum* (Haeckel, 1888) | - | - | 0.5 ± 0.9 | 0.3–3.9 | 27.6 |
| *Lensia meteori* (Leloup, 1934) | - | - | 1 ± 2.3 | 0.3–14.9 | 25.3 |
| *Lychnagalma utricularia* (Claus, 1879) | 0.7 ± 2.37 | 3.6–10.2 | 0.5 ± 1.2 | 0.5–7.4 | 25.3 |
| *Hippopodius hippopus* (Forsskål, 1776) | - | - | 0.3 ± 0.7 | 0.1–2.8 | 18.4 |
| *Amphicaryon* sp. | - | - | 0.3 ± 0.8 | 0.4–3.5 | 14.9 |
| *Halistemma rubrum* (Vogt, 1852) | - | - | 0.2 ± 0.5 | 0.4–1.9 | 14.9 |
| *Lensia subtilis* (Chun, 1886) | - | - | 0.2 ± 0.7 | 0.1–3.4 | 11.5 |
| *Lensia* spp. | - | - | 0.2 ± 0.8 | 0.6–4.6 | 8 |
| *Muggiaea kochii* (Will, 1844) | 95.7 ± 282.9 | 70.04–1185.4 | 0.2 ± 1.3 | 1.9–9.1 | 8 |
| *Athorybia rosacea* (Forsskål, 1775) | 0.1 ± 0.4 | 1.6–1.6 | 0.1 ± 0.7 | 0.4–4.7 | 6.9 |
| *Rosacea plicata* Bigelow, 1911 | - | - | 0.1 ± 0.5 | 0.6–2.5 | 6.9 |
| *Sulculeolaria monoica* (Chun, 1888) | - | - | 0.1 ± 0.5 | 0.5–2.4 | 6.9 |
| *Enneagonum hyalinum* Quoy & Gaimard, 1827 | 207.1 ± 589.1 | 152.8–2227.1 | 0.1 ± 0.2 | 1.4–1.4 | 5.7 |
| *Chuniphyes* sp. | - | - | 0.1 ± 0.5 | 0.1–4.2 | 4.6 |
| *Lensia cossack* Totton, 1941 | - | - | 0.1 ± 0.1 | 0.4–0.6 | 4.6 |
| *Lensia fowleri* (Bigelow, 1911) | - | - | 0.1 ± 0.3 | 0.3–1.6 | 4.6 |
| *Abyla* sp. | - | - | 0.1 ± 0.2 | 0.1–1.1 | 3.4 |
| *Abyla trigona* Quoy & Gaimard, 1827 | - | - | 0.1 ± 0.3 | 0.8–1.6 | 3.4 |
| *Forskalia edwardsii* Kölliker, 1853 | - | - | 0.1 ± 0.3 | 1.2–1.5 | 3.4 |
| *Lensia conoidea* (Keferstein & Ehlers, 1860) | - | - | 0.1 ± 0.2 | 0.5–1.5 | 3.4 |
| *Lensia hardy* Totton, 1941 | - | - | 0.1 ± 0.9 | 0.7–6.5 | 3.4 |
| *Sphaeronectes koellikeri* Huxley, 1859 | - | - | 0.1 ± 0.3 | 0.9–1.9 | 3.4 |
| *Forskalia contorta* (Milne Edwards, 1841) | - | - | 0.1 ± 0.3 | 0.5–2.1 | 2.3 |
| *Lensia subtiloides* (Lens & van Riemsdijk, 1908) | - | - | 0.1 ± 0.1 | 0.3–0.3 | 2.3 |
| *Amphicaryon peltifera* (Haeckel, 1888) | - | - | 0.1 ± 0.28 | 2.3–2.3 | 1.1 |
| *Dimophyes arctica* (Chun, 1897) | - | - | 0.1 ± 0.76 | 6.2–6.2 | 1.1 |
| *Lensia hotspur* Totton, 1941 | - | - | 0.1 ± 1 | 8.2–8.2 | 1.1 |

*(Continued)*

**Table 1.** (Continued)

| Species | Neritic | | Oceanic | | f |
|---|---|---|---|---|---|
| | Mean ± SD | Range of non-zero abundances | Mean ± SD | Range of non-zero abundances | |
| *Lensia leloupi* Totton, 1954 | - | - | 0.1 ± 0.2 | 1.9–1.9 | 1.1 |
| *Physophora hydrostatica* Forsskål, 1775 | - | - | 0.1 ± 0.3 | 2.5–2.4 | 1.1 |
| *Rosacea* sp. | - | - | 0.1 ± 0.1 | 1.2–1.2 | 1.1 |
| *Sulculeolaria quadrivalvis* de Blainville, 1830 | - | - | 0.1 ± 0.1 | 0.1–0.1 | 1.1 |
| **Hydromedusae** | | | | | |
| *Liriope tetraphylla* (Chamisso & Eysenhardt, 1821) | 322.1 ± 582.2 | 2.8–2199.6 | 9.2 ± 14.6 | 0.3–99.3 | 88.5 |
| *Aglaura hemistoma* Péron & Lesueur, 1810 | 10.3 ± 31.3 | 11.6–125.9 | 33 ± 37.6 | 0.1–192.7 | 78.2 |
| *Sminthea eurygaster* Gegenbaur, 1857 | - | - | 2.5 ± 4 | 0.2–16.7 | 41.4 |
| *Rhopalonema velatum* Gegenbaur, 1857 | 0.6 ± 2.6 | 11.5–11.4 | 1.1 ± 1.9 | 0.1–7.9 | 35.6 |
| *Solmundella bitentaculata* (Quoy & Gaimard, 1833) | 0.6 ± 1.7 | 1.6–7.2 | 1.1 ± 2 | 0.5–9.3 | 34.5 |
| *Cytaeis* sp.1 | 0.4 ± 1.7 | 7.6–7.6 | 0.5 ± 1 | 0.3–5.5 | 29.9 |
| *Annatiara affinis* (Hartlaub, 1914) | 0.4 ± 1.7 | 7.6–7.6 | 0.4 ± 1.5 | 0.6–10.6 | 12.6 |
| *Clytia* spp. | 0.5 ± 1.2 | 3.5–3.5 | 0.1 ± 0.4 | 0.5–1.9 | 11.5 |
| *Cunina octonaria* McCrady, 1859 | 5.6 ± 17.2 | 4.8–75.4 | 0.1 ± 0.7 | 0.5–4.5 | 9.2 |
| *Cirrholovenia tetranema* Kramp, 1959 | 2.5 ± 7.7 | 2.4–34.3 | 0.1 ± 0.5 | 1–3 | 8 |
| *Persa incolorata* McCrady, 1859 | 222.7 ± 646.7 | 3.6–2701.3 | 0.2 ± 1.5 | 12–12.0 | 6.9 |
| *Eucheilota maculata* Hartlaub, 1894 | 0.6 ± 2.4 | 1.5–10.5 | 0.1 ± 0.4 | 0.9–2.27 | 5.7 |
| *Cirrholovenia polynema* Kramp, 1959 | - | - | 0.1 ± 0.2 | 0.4–1.07 | 4.6 |
| *Aequorea* spp. | - | - | 0.1 ± 0.2 | 0.9–1.2 | 3.4 |
| *Cunina frugifera* Kramp, 1948 | - | - | 0.1 ± 0.1 | 1–2.7 | 3.4 |
| *Helgicirrha angelicae* Tosetto, Neumann-Leitão & Nogueira Junior, 2020 | 9.6 ± 39.3 | 4.8–176 | - | - | 3.4 |
| Anthomedusa sp.1 | - | - | 0.1 ± 0.6 | 1.7–4.7 | 2.3 |
| Anthomedusa sp.3 | - | - | 0.1 ± 0.7 | 1.9–5.1 | 2.3 |
| *Eucheilota* spp. | 0.1 ± 0.14 | 0.6–0.6 | 0.1 ± 0.2 | 1.9–1.9 | 2.3 |
| *Eutima marajoara* Tosetto, Neumann-Leitão & Nogueira Junior, 2020 | 13.4 ± 45.5 | 76.4–192.6 | - | - | 2.3 |
| Levenellidae sp. | - | - | 0.1 ± 0.2 | 0.5–1.7 | 2.3 |
| *Malagazzia carolinae* (Mayer, 1900) | 0.2 ± 0.8 | 0.8–3.6 | - | - | 2.3 |
| Bougainvilliidae sp | 2.9 ± 10.0 | 14.9–43.1 | - | - | 2.3 |
| *Obelia* sp. | 3.1 ± 13.8 | 0.6–61.8 | - | - | 2.3 |
| *Pegantha martagon* Haeckel, 1879 | - | - | 0.1 ± 0.1 | 0.4–0.6 | 2.3 |
| *Aequorea forskalea* Péron & Lesueur, 1810 | - | - | 0.1 ± 0.2 | 1.9–1.9 | 1.1 |
| *Stomotoca atra* L. Agassiz, 1862 | 0.1 ± 0.2 | 0.8–0.8 | - | - | 1.1 |
| Anthomedusa sp.2 | - | - | 0.1 ± 0.2 | 1.9–1.9 | 1.1 |
| Anthomedusa sp.4 | - | - | 0.1 ± 0.3 | 2.2–2.2 | 1.1 |
| *Bougainvillia muscus* (Allman, 1863) | - | - | 0.1 ± 0.1 | 0.5–0.5 | 1.1 |
| Campanulariidae sp. | - | - | 0.1 ± 0.3 | 2.5–2.5 | 1.1 |
| Corynidae sp. | - | - | 0.1 ± 0.1 | 0.5–0.5 | 1.1 |
| *Cytaeis* sp.2 | - | - | 0.1 ± 0.1 | 0.9–0.9 | 1.1 |
| *Eirene lactea* (Mayer, 1900) | 0.2 ± 0.8 | 3.5–3.5 | - | - | 1.1 |
| Hydromedusae sp. | - | - | 0.1 ± 0.1 | 0.6–0.6 | 1.1 |
| *Laodicea undulata* (Forbes & Goodsir, 1853) | 1.1 ± 5.1 | 22.9–22.9 | - | - | 1.1 |
| *Mitrocomium cirratum* Haeckel, 1879 | 0.1 ± 0.4 | 1.6–1.6 | - | - | 1.1 |
| *Octophialucium bigelowi* Kramp, 1955 | 0.2 ± 0.7 | 3.2–3.2 | - | - | 1.1 |
| *Octophialucium haeckeli* (Vannucci & Soares Moreira, 1966) | 0.2 ± 0.8 | 3.5–3.5 | - | - | 1.1 |

(*Continued*)

**Table 1.** (Continued)

| Species | Neritic | | Oceanic | | f |
|---|---|---|---|---|---|
| | Mean ± SD | Range of non-zero abundances | Mean ± SD | Range of non-zero abundances | |
| *Olindias* sp. | 0.1 ± 0.4 | 1.59–1.6 | - | - | 1.1 |
| *Pegantha triloba* Haeckel, 1879 | - | - | 0.1 ± 0.3 | 2.8–2.8 | 1.1 |
| **Scyphomedusae** | | | | | |
| *Nausithoe punctata* Kölliker, 1853 | 0.6 ± 2.3 | 1.49–10.3 | 0.3 ± 0.7 | 0.3–3.3 | 20.7 |
| *Nausithoe maculata* Jarms, 1990 | - | - | 0.1 ± 0.2 | 1.7–1.7 | 1.1 |
| **Other** | | | | | |
| Cerinula larvae | 8.9 ± 19.7 | 1.49–82.6 | 0.4 ± 0.9 | 0.2–4.6 | 29.9 |
| Ephyrae larvae | 0.5 ± 1.5 | 1.43–6.1 | 0.1 ± 0.3 | 0.4–1.8 | 8 |
| Athorybia larvae | - | - | 0.1 ± 0.6 | 0.3–4.9 | 2 |

Group C, that occurred mainly in oceanic stations, encompassed three subgroups and two outliers. Stations 45 and 10 were located near the shelf break and considered outliers. Subgroup C1 clustered stations 34 to 38, located in the southeastern portion of the study area (Fig 7). It was mainly represented by *C. appendiculata*, *B. bassensis* and *A. tetragona* and differed from other oceanic groups by the low occurrences of *E. mitra* and *S. chuni* (Fig 6; Table 2). The similarity within the group was 69.2%. Subgroup C2 included the remaining oceanic stations outside the influence of the ARP. With an average similarity of 69.5%, *C. appendiculata*, *E. mitra* and *B. bassensis* were the main representative of this subgroup (Fig 7; Table 3). Except for station 17 and 22 placed in subgroup C2, all oceanic stations under the influence of the ARP and stations 25 and 26 located near its limit were included in subgroup C3 (Fig 7; Table 3), with 69.3% of similarity. This subgroup was mainly represented by high abundances of *D. bojani*, *E. mitra* and *A. hemistoma*.

### Responses to mesoscale processes and environmental gradients

The first two axes of the RDA explained 54.9% of planktonic cnidarian species variance (Table 4). The Monte Carlo test showed that the first (F-ratio = 18.3, P-value = 0.002) and all canonical axes together (F-ratio = 6.3, P-value = 0.002) were significant. Axis 1 (37.8% of variance) was mainly related to the oceanic/neritic gradient. Zooplankton biomass and cold-core cyclonic eddies were positively related to this axis. The second axis (17% of variance) was negatively related to surface salinity and cyclonic eddies, and positively related to the ARP, fluorescence and zooplankton biomass. Axes 3 and 4 explained together less than 10% of species variance and were not considered (Fig 8; Table 4).

Most oceanic species were closely related to the left portion of the first axis with few relation with axis 2, which represented the ARP and surface salinity gradient, reflecting their wide distribution over the oceanic province and low effect of the ARP on their distribution. Other oceanic species, such as *D. bojani*, *E. mitra* and *R. velatum* correlated with both the negative portion of axis 1 and the positive portion of axis 2, indicating their higher abundance in the oceanic environment under influence of the ARP (Fig 8).

*Nanomia bijuga*, *S. bitentaculata* and *L. campanella* were close related with the positive portion of axis two, indicating their preference for the low salinity environment of the ARP in both oceanic and neritic habitats. *Enneagonum hyalinum*, *M. kochii*, *P. incolorata* and *D. dispar* correlated with the positive portions of axes 1 and 2, reflecting their high abundances in neritic stations under the ARP influence, where the higher primary production and food availability occurred. *Liriope tetraphylla* was positively related to the first axis, as a result of its large

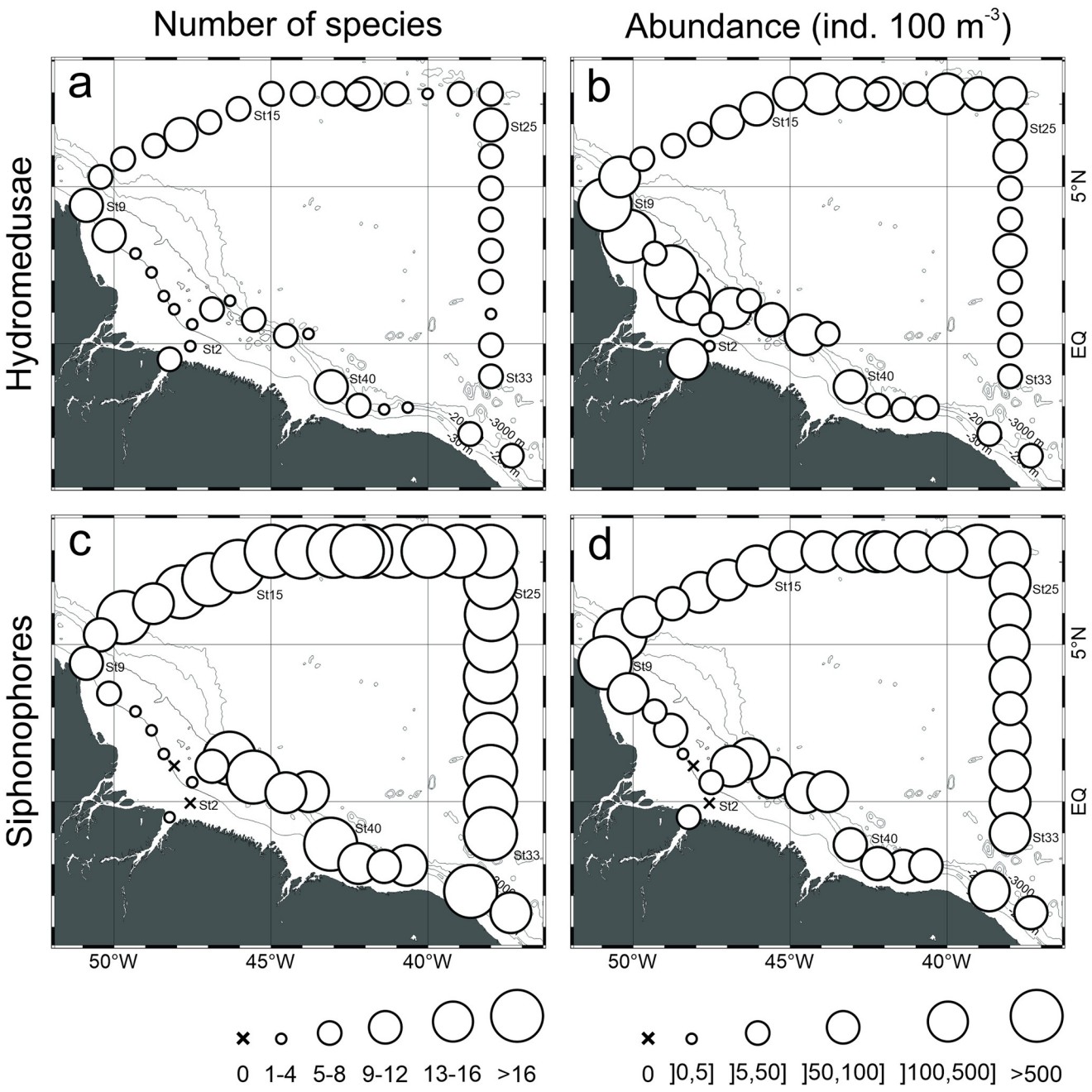

**Fig 4. Geographic distribution of species richness and total abundance of hydromedusae and siphonophores found in the top 200 m of the water column (mean of samples collected with 120 and 300 μm mesh nets) in October 2012.**

abundance over the continental shelf both inside and outside the ARP and in the cold-core cyclonic eddy (Fig 8).

## Discussion

Our results show that the Amazon River Plume drives physical and ecological processes that affect the distribution and abundance of planktonic cnidarians both over the continental shelf

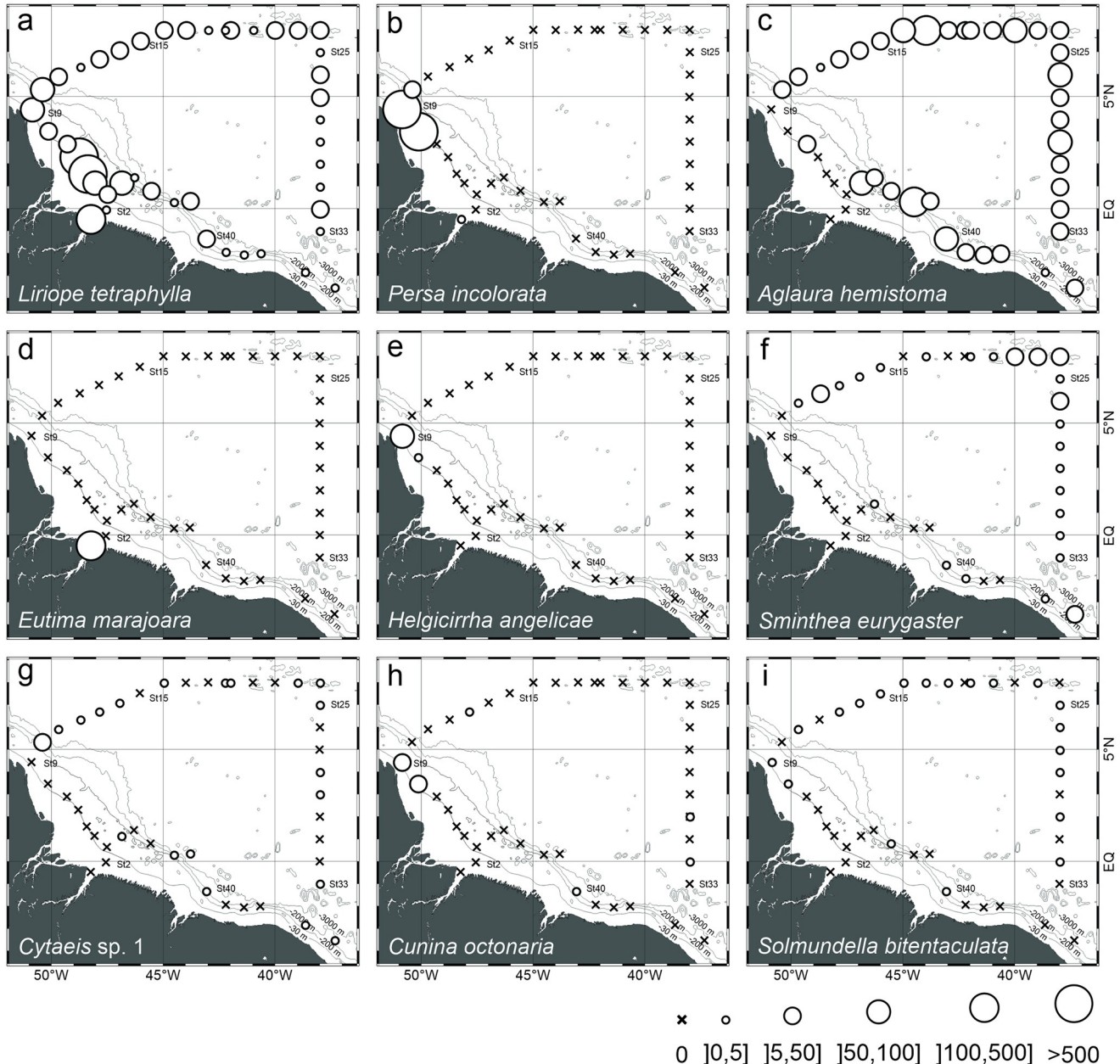

**Fig 5. Geographic distribution and abundance of the dominant hydromedusae found in the top 200 m of the water column (mean of samples collected with 120 and 300 μm mesh nets) in October 2012.** Species are sorted according to total abundance.

and in the open ocean in the Western Equatorial Atlantic Ocean. Although our study was focused on cnidarians, analogous effects are expected to the whole planktonic community [e.g. 28]. Over the continental shelf, we observed a strong influence of the ARP in the west of the study domain. This is the zone where the NBC is retroflected to the north, spreading the ARP over the entire length of the shelf [58]. In addition to the freshwater input lowering salinity, the ARP discharges a massive amount of nutrients, organic matter and sediments [24], boosting the primary production. Consequently, higher trophic levels, including planktonic cnidarians benefit from the higher food availability derived from bottom-up processes.

**Table 2. Results of the analysis of variance testing differences in the abundance of representative planktonic cnidarian species among neritic and oceanic provinces inside (ARP) and outside (Out ARP) the influence of the Amazon River Plume and Tukey post-hoc test.** Significant p-values ($< 0.05$) are in bold. Different Letters indicate significant pair-wise difference among areas in the Tukey test. Species are sorted according to total abundance.

| Species | F | p | Neritic | | Oceanic | |
|---|---|---|---|---|---|---|
| | | | Out ARP | ARP | Out ARP | ARP |
| *Liriope tetraphylla* | 7.92 | **0.000** | a | b | c | c |
| *Persa incolorata* | 311.87 | **0.000** | a | b | a | a |
| *Enneagonum hyalinum* | 318.16 | **0.000** | a | b | a | a |
| *Diphyes bojani* | 15.35 | **0.000** | a | a | b | c |
| *Chelophyes appendiculata* | 19.26 | **0.000** | a | a | b | b |
| *Aglaura hemistoma* | 13.89 | **0.000** | a | a | b | b |
| *Eudoxoides mitra* | 19.07 | **0.000** | a | a | b | c |
| *Muggiaea kochii* | 110.53 | **0.000** | a | b | a | a |
| *Diphyes dispar* | 12.21 | **0.000** | a | b | a | a |
| *Bassia bassensis* | 17.41 | **0.000** | a | a | b | b |
| *Eudoxoides spiralis* | 9.94 | **0.000** | a | a | b | c |
| *Sulculeolaria chuni* | 3.81 | **0.017** | a | a | b | b |
| *Abylopsis tetragona* | 10.93 | **0.000** | a | a | b | b |
| *Abylopsis eschscholtzii* | 5.23 | **0.004** | a | a | b | b |
| *Nanomia bijuga* | 7.68 | **0.000** | a | b | c | d |

Over the continental shelf, the ARP was restricted to the surface layer. Below lied an oxygen minimum layer, likely in consequence of the high rates of organic matter that sink and fuel microbial respiration [59]. These features compressed the bulk of productivity in the ARP, both vertically and horizontally (Fig 9). Although with the oblique trawls performed in our study, we could not infer the exact vertical position of the huge amount of cnidarians observed in the area of influence of the ARP and its relation with the oxygen minimum layer, such three-dimensional habitat configuration seems particularly beneficial for cnidarians. While cnidarians usually are not restricted by oxygen concentration due their low metabolic demand, low oxygen levels compress the suitable habitat for fish and other predators with higher metabolic demand to the thin surface layer [32,60–62]. In addition, in this thin productive layer the water has low visibility due to the sediment runoff and suspended particulate matter. Cnidarians, which are not visual feeders, can outcompete visual predators and, due their fast reproduction, proliferate in large population blooms [63–65]. Thus, in the habitat shaped by the ARP, we observed the highest abundance of planktonic cnidarians and high species richness for both hydromedusae and siphonophores. Such huge abundances are in general higher than the observed over the continental shelf in other tropical systems along the Western Atlantic [21,66] and the maxima observed in high productive scenarios, such as upwelling and river runoff, in other western boundary systems [20,67].

In relation to the biodiversity in this system, among several species that were abundant over the continental shelf under influence of the ARP, the hydromedusae *P. inclorata*, *C. octonaria* and *H. angelicae*, and the siphonophores *E. hyalinum* and *M. kochii* occurred almost exclusively within this habitat. *H. angelicae* was recently described from the study area [68], perhaps being an endemic representative of the ARP fauna. *M. kochii* and *C. octonaria* are typical coastal water species in the Western Atlantic [5,21,39,69], therefore its absence in the remaining habitats of the study area are expected. Distinctly, *P. inclorata* and *E. hyalinum* have been reported with contrasting niches in different works. In some of them, similar to our results, they were reported as coastal water species [33,35,70,71] and others as high saline oceanic species [30,72]. Such differences can represent intraspecific variability or even cryptic species.

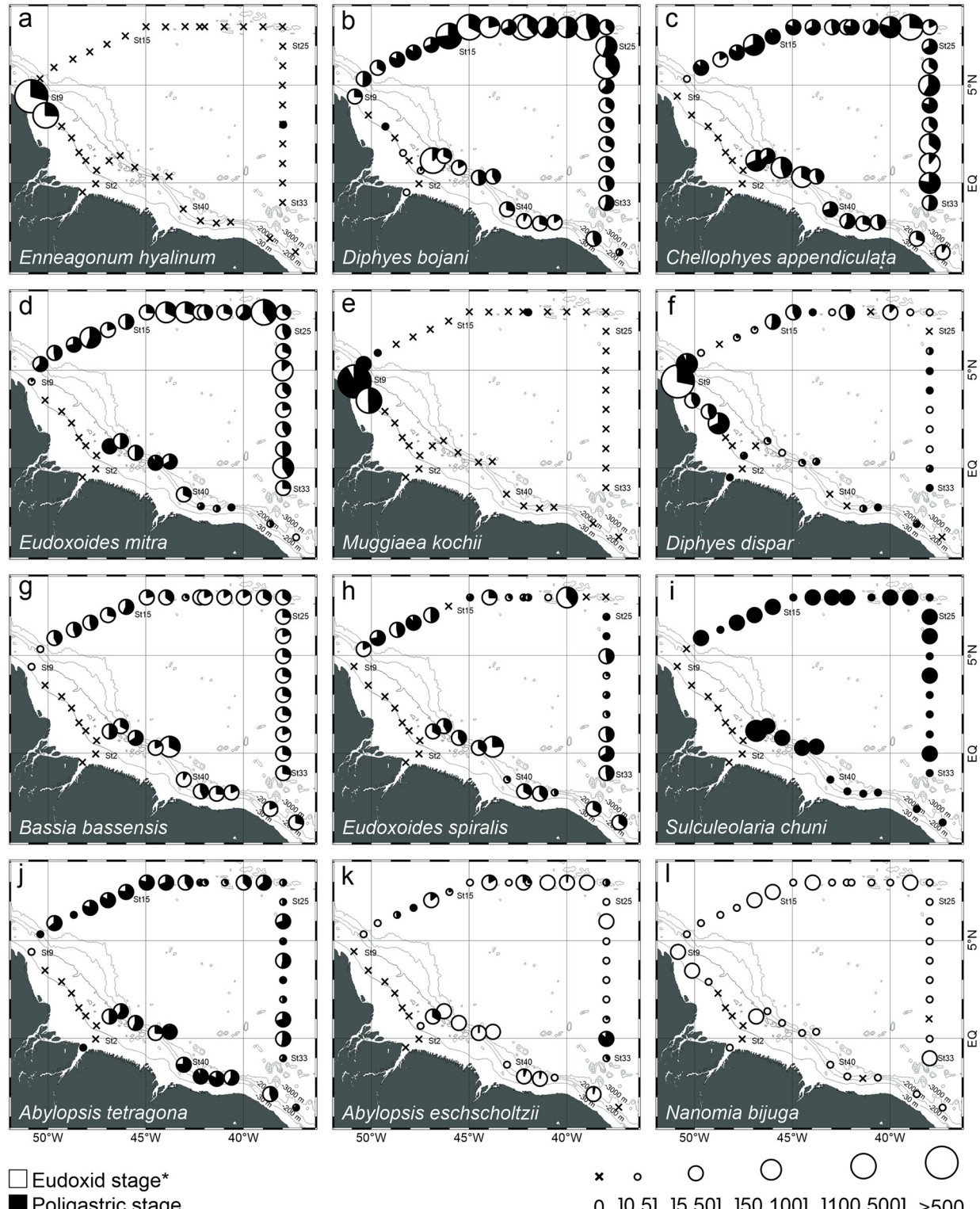

**Fig 6. Geographic distribution and abundance of the dominant siphonophores found in the top 200 m of the water column (mean of 120 and 300 μm meshes) in October 2012.** *Except for *Nanomia bijuga*. Species are sorted according to total abundance.

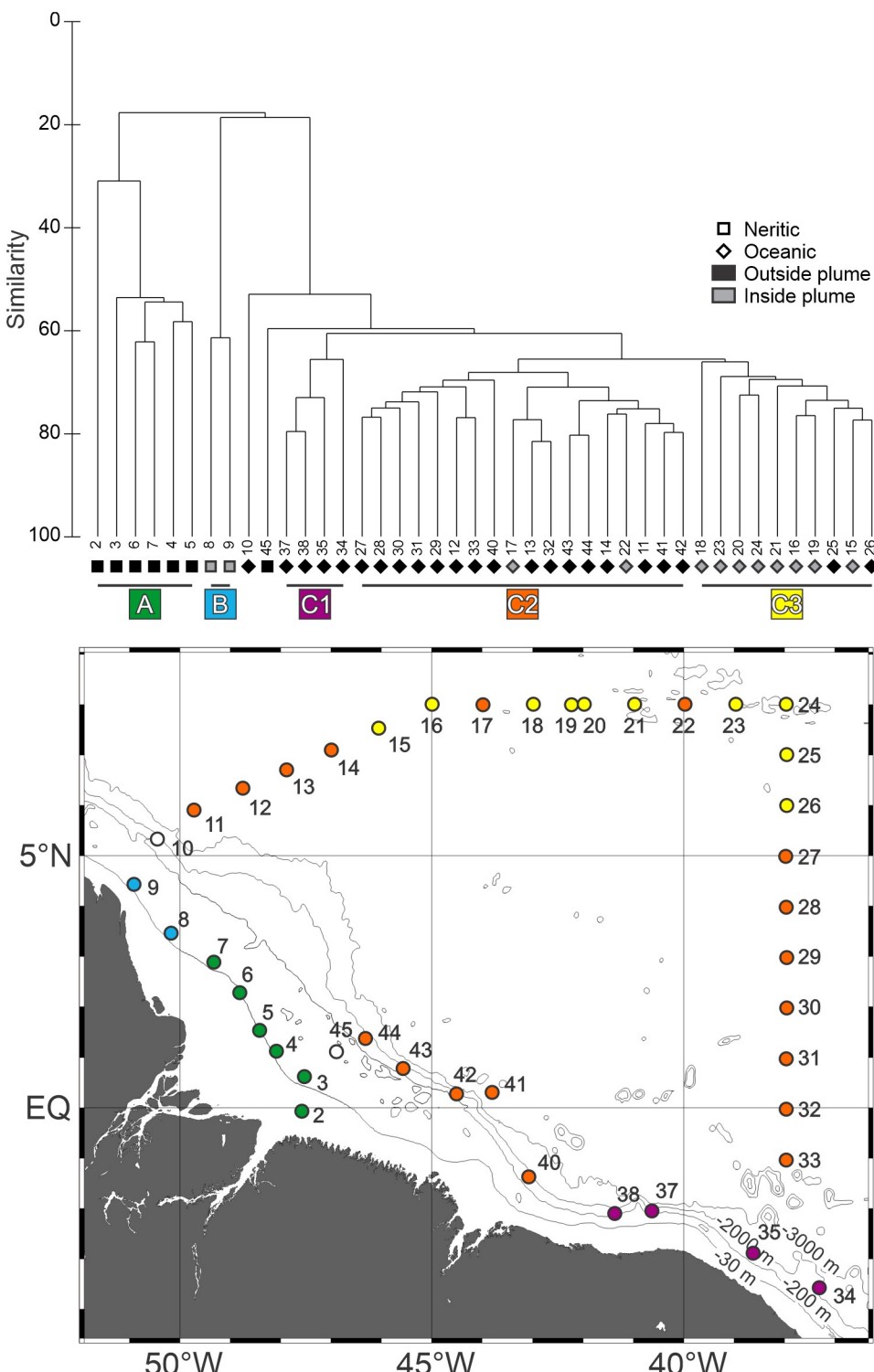

**Fig 7. Cluster analysis dendogram indicating five groups of stations with similar planktonic cnidarian communities in the Western Equatorial Atlantic Ocean.**

**Table 3. Results of Similarity Percentage analysis (SIMPER), showing the relative contribution of planktonic cnidarian species in the formation of the groups defined in the Cluster analysis.**

| Species | A | B | C1 | C2 | C3 |
|---|---|---|---|---|---|
| *Abylopsis eschscholtzi* | - | - | 5.2 | 4.2 | 4.4 |
| *Abylopsis tetragona* | - | - | 11.9 | 6.9 | 4.6 |
| *Agalma okeni* | - | - | - | 2.1 | 2 |
| *Aglaura hemistoma* | - | - | 11.2 | 9.9 | 10.7 |
| *Bassia bassensis* | - | - | 17.1 | 10.1 | 7.9 |
| *Ceratocymba leuckartii* | - | - | - | - | 2.1 |
| *Chelophyes appendiculata* | - | - | 18.8 | 13.4 | 8.2 |
| *Cunina octonaria* | - | 6.8 | - | - | - |
| *Diphyes bojani* | - | - | 10.8 | 9.9 | 13.1 |
| *Diphyes dispar* | 8.1 | 11.5 | - | - | - |
| *Enneagonum hyalinum* | - | 16.3 | - | - | - |
| *Eudoxoides mitra* | - | - | - | 10.8 | 11.2 |
| *Eudoxoides spiralis* | - | - | 11.3 | 7.7 | - |
| Cerinula larvae | - | 10.1 | - | - | - |
| *Lensia campanella* | - | - | - | - | 2.7 |
| *Liriope tetraphylla* | 84.1 | 5.7 | 5 | 5.2 | 5.1 |
| *Muggiaea kochi* | - | 14.6 | - | - | - |
| *Nanomia bijuga* | - | 6.7 | - | 2.4 | 3.2 |
| *Persa incolorata* | - | 21.3 | - | - | - |
| *Rhopalonema velatum* | - | - | - | - | 2.1 |
| *Sminthea eurygaster* | - | - | - | 1.8 | 1.9 |
| *Solmundella bitentaculata* | - | - | - | - | 1.6 |
| *Sulculeolaria biloba* | - | - | - | - | 2.6 |
| *Sulculeolaria chuni* | - | - | - | 6.2 | 4.9 |
| *Sulculeolaria turgida* | - | - | - | - | 2.3 |

In the portion of the continental shelf outside the influence of the ARP, as a typical western boundary system where strong coastward currents carry oligotrophic oceanic waters over the shelf, we observed high salinity oligotrophic waters all along its extension. Under such conditions, characteristically oceanic species were expected to be present and dominant over the continental shelf [20,21]. Instead, the holoplanktonic hydromedusa *L. tetraphylla* dominated almost alone and with quite high abundance, while only occasional catches of other species were observed. *L. tetraphylla* is abundant and widely distributed in neritic habitats of the Western Atlantic [5,21,39,66,73]. However, unlike our results, most studies reported the species co-occurring with other dominant species in both purely coastal assemblages and/or coastal and open ocean mixed assemblages. The continental shelf outside the Amazon basin is generally wider and more complex, with several eddies, internal waves and high tide amplitude, than the remaining Tropical Western Atlantic [19,58,74]. Such characteristics are likely behind the processes differencing the cnidarian assemblage between the Amazonian shelf outside the influence of the ARP and other analogous systems.

In the open ocean we observed the influence of the ARP along the NECC, which is derived from the NBC retroflexion and seasonally spreads the ARP to the Central Atlantic [43]. Distinctly from the continental shelf, in this system we did not observe contrasting differences in surface fluorescence when comparing to the remaining open ocean. However, similar to the continental shelf, we indeed observed an oxygen minimum layer below the ARP waters

**Table 4. Summary of the Redundancy Analysis (RDA) performed between the cnidarian assemblage and environmental explanatory variables from the Western Equatorial Atlantic Ocean.**

|  | Axis 1 | Axis 2 |
|---|---|---|
| Eigenvalues | 0.378 | 0.17 |
| Species-environment correlations | 0.912 | 0.95 |
| **Cumulative variance (%):** |  |  |
| Of species data | 37.8 | 54.9 |
| Of species-environment relationships | 54.1 | 78.5 |
| **Correlations of explanatory variables:** |  |  |
| Neritic province | 0.8967 | -0.1822 |
| Oceanic province | -0.8967 | 0.1822 |
| Amazon River Plume | -0.005 | 0.6031 |
| North Brazil Current | 0.4703 | -0.3454 |
| North Equatorial Counter Current | -0.4377 | 0.3145 |
| Anticyclonic eddy | 0.0142 | 0.283 |
| Cyclonic eddy | 0.529 | -0.476 |
| Sea surface temperature | -0.0872 | 0.2824 |
| Sea surface salinity | -0.1898 | -0.8106 |
| Dissolved Oxygen | -0.4794 | 0.4573 |
| Fluorescence | 0.2873 | 0.7395 |
| Zooplankton Biomass | 0.5882 | 0.3632 |

(50~100 m depth). In accordance, the differences in the oceanic cnidarian assemblage inside and outside the influence of the ARP were less preeminent than the observed over the continental shelf, but still notable. The same species were dominant and a similar number of species was observed in the open ocean inside and outside the influence of the ARP. Most of these species are holoplanktonic, and without a benthic polypoid stages that could require substrata for settlement in their life cycle [75,76], they are able to disperse freely. Thus they are typically present in tropical oceanic waters from the Western Atlantic [21,77]. Apparently, the low salinity observed inside the oceanic ARP, did not restricted the distribution of these dominant species there. However, since our oblique zooplankton samples were performed from 200 meters depth to surface, and the low salinity ARP waters did not exceed 50 meters depth, species intolerant to lower salinity could be distributed below the ARP. A study considering stratified samples is necessary to clarify this issue properly. In any case, some of these species may be occasionally found in low salinities inside estuaries in the Western Atlantic [78–80].

Despite the absence of exclusive dominant species, differences in species abundance were perceptible between the habitats inside and outside the ARP in the open ocean. Although we did observed strong differences in fluorescence between both habitats, which is an indicator of primary productivity, higher biomass of cyanobacteria (primary producers) [81] and total zooplankton (food for cnidarians) were observed along the oceanic region under influence of the ARP. Such differences along the food web, the habitat compression caused by the oxygen minimum zone bellow the ARP and the reduced salinity (Fig 9) are likely responsible for the differences we observed in the planktonic cnidarian assemblage abundance in the open ocean.

## Conclusions

This was the first detailed survey on the cnidarian assemblage structure from a major river plume reaching mesoscale dimensions in neritic and oceanic provinces. Our results showed that the freshwater, nutrients and sediment runoff of the Amazon River shapes habitats with a

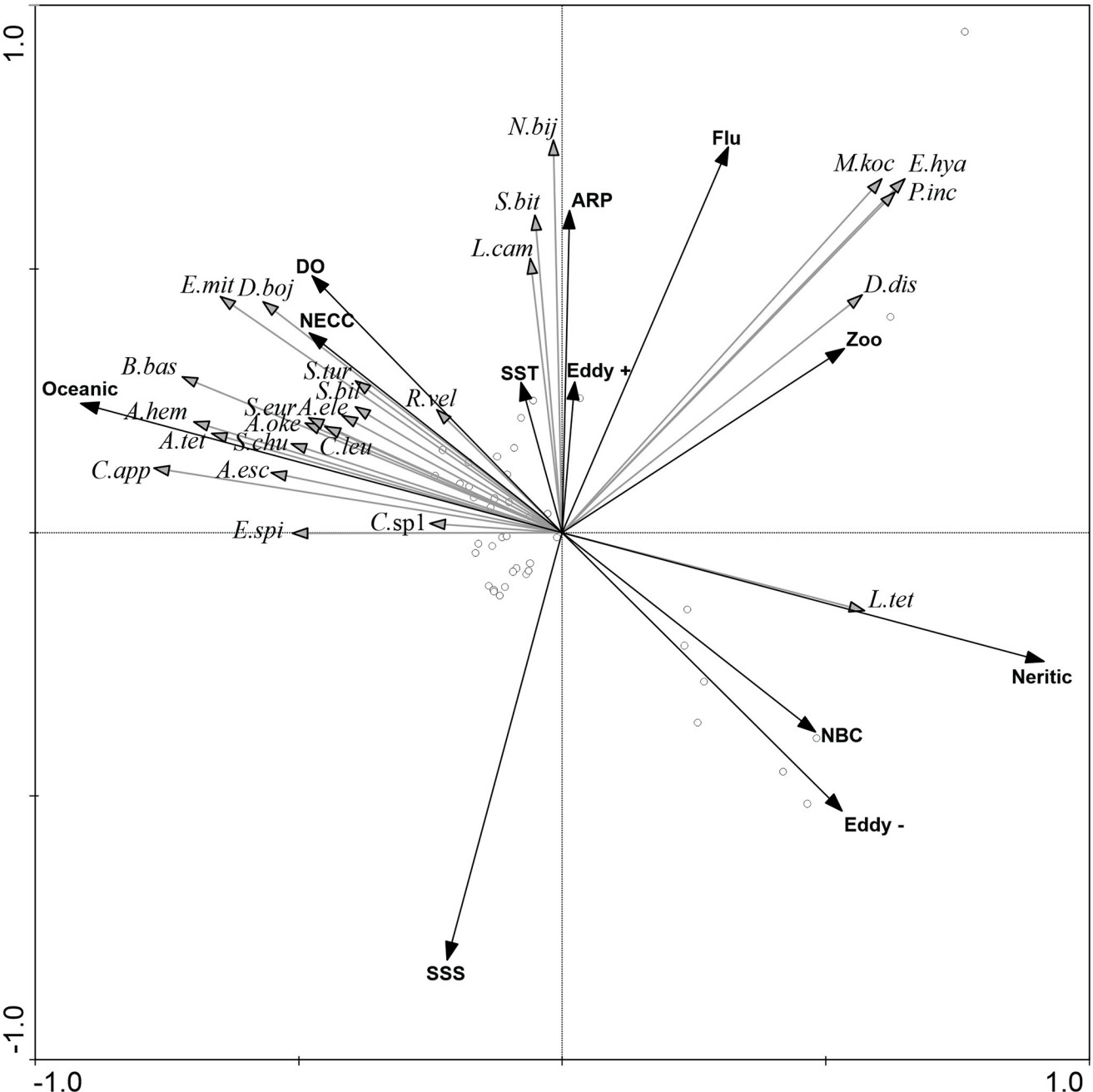

**Fig 8. Redundancy analysis relating dominant planktonic cnidarian species to environmental gradients and mesoscale processes in the Western Equatorial Atlantic Ocean.** *L.tet = Liriope tetraphylla, D.boj = Diphyes bojani, A.hem = Aglaura hemistoma, E.mit = Eudoxoides mitra, B.bas = Bassia bassensis, A.tet = Abylopsis tetragona, N.bij = Nanomia bijuga, C.app = Chellophyes appendiculata, A.esc = Abylopsis eschscholtzii, S.chu = Sulculeolaria chuni, D.dis = Diphyes dispar, A.oke = Agalma okeni, E.spi = Eudoxoides spiralis, S.eur = Sminthea eurygaster, C.leu = Ceratocymba leuckartii, S. tur = Sulculeolaria turgida, R.vel = Rhopalonema velatum, L.cam = Lensia campanella, A.ele = Agalma elegans, C.sp1 = Cytaeis sp. 1, S.bit = Solmundella bitentaculata, S.bil = Sulculeolaria biloba, P.inc = Persa incolorata, E.hya = Enneagonum hyalinum, M.koc = Muggiaea kochii,* ARP = Presence of Amazon River Plume, NBC = North Brazilian Current, NECC = North Equatorial Countercurrent, Eddy + = Anticyclonic eddy, Eddy— = Cyclonic eddy, SST = Sea surface temperature, SSS = Sea surface salinity, DO = Dissolved oxygen, Flu = Fluorescence, Zoo = Zooplankton Biomass.

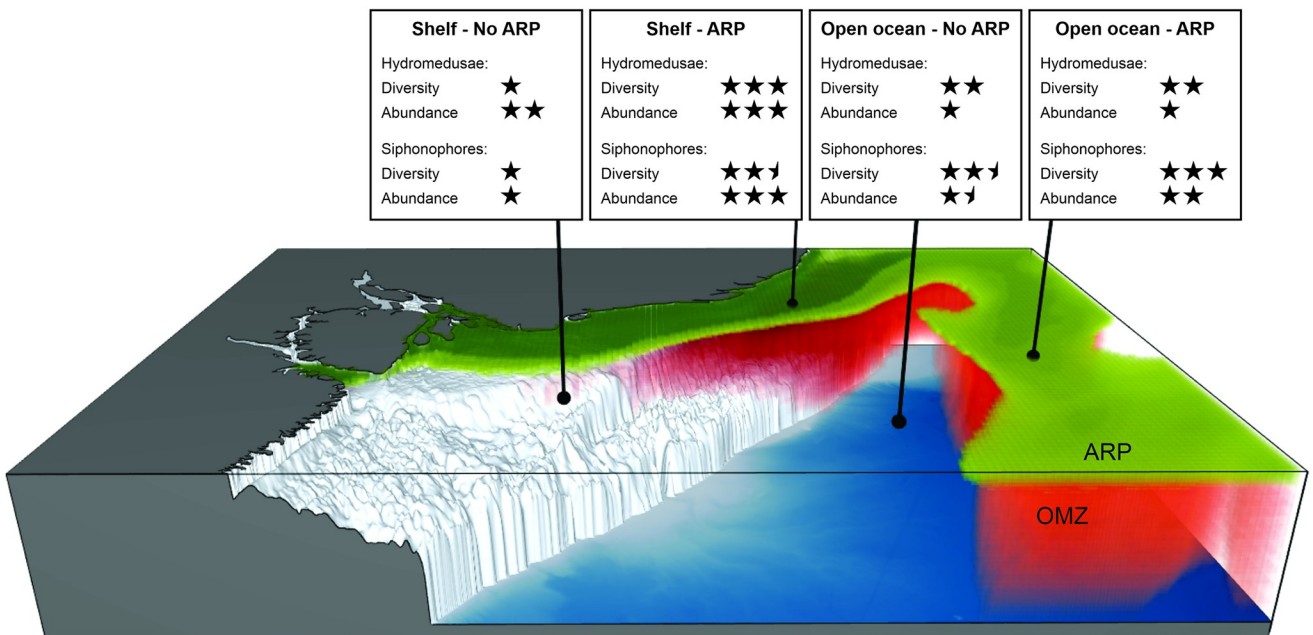

**Fig 9. Schematic representation of the Amazon River Plume (ARP) and associated oxygen minimum zone (OMZ) and effects of habitat structure on planktonic cnidarian assemblages in the Equatorial Western Atlantic Ocean.**

thin surface highly productive system compressed by a deeper oxygen minimum zone both over the continental shelf and in the open ocean in the Western Equatorial Atlantic Ocean. We hypothesized that such habitat structure is particularly advantageous to planktonic cnidarians, which have low metabolic rates and are able to survive in hypoxic zones, resulting in high species richness and abundance. As expected, the Western Equatorial Atlantic Ocean under the influence of the ARP revealed a complex system, with many physical and biogeochemical processes occurring simultaneously. This complexity was reflected in the structure of cnidarian assemblage.

## Acknowledgments

We are grateful to the officers, crew and scientific team of the Camadas Finas III and Amadeus II research project for their contribution to the success of the operations. The present study was not possible without the support of all members from LABZOO and other laboratories from UFPE and UFRPE. We are grateful to the Brazilian National Institute of Science and Technology for Tropical Marine Environments (INCT AmbTropic), the Brazilian Research Network on Global Climate Change (Rede CLIMA) and European Integrated CARBO-CHANGE and all the scientific team on board of the Camadas Finas III cruise. We thank the CNPq (Brazilian National Council for Scientific and Technological Development). This work is a contribution to the LMI TAPIOCA (www.tapioca.ird.fr), CAPES/COFECUB program, the European Union's Horizon 2020 projects PADDLE and TRIATLAS.

## Author Contributions

**Conceptualization:** Everton Giachini Tosetto, Sigrid Neumann-Leitão, Moacyr Araujo, Miodeli Nogueira Júnior.

**Data curation:** Sigrid Neumann-Leitão, Moacyr Araujo.

**Formal analysis:** Everton Giachini Tosetto, Djoirka Minto Dimoune, Arnaud Bertrand, Miodeli Nogueira Júnior.

**Funding acquisition:** Moacyr Araujo.

**Investigation:** Everton Giachini Tosetto, Miodeli Nogueira Júnior.

**Methodology:** Everton Giachini Tosetto, Sigrid Neumann-Leitão, Moacyr Araujo, Djoirka Minto Dimoune, Arnaud Bertrand, Miodeli Nogueira Júnior.

**Project administration:** Everton Giachini Tosetto, Sigrid Neumann-Leitão, Miodeli Nogueira Júnior.

**Resources:** Sigrid Neumann-Leitão, Moacyr Araujo.

**Supervision:** Miodeli Nogueira Júnior.

**Validation:** Everton Giachini Tosetto.

**Visualization:** Everton Giachini Tosetto, Sigrid Neumann-Leitão.

**Writing – original draft:** Everton Giachini Tosetto.

**Writing – review & editing:** Everton Giachini Tosetto, Sigrid Neumann-Leitão, Moacyr Araujo, Djoirka Minto Dimoune, Arnaud Bertrand, Miodeli Nogueira Júnior.

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
