## [Decision Letter · Decision Letter 0]

14 Jun 2023

PONE-D-23-10398Amazon River plume habitats shape planktonic cnidarian assemblages in the Western AtlanticPLOS ONE

Dear Dr. Tosetto,

Thank you for submitting your manuscript to PLOS ONE. After careful consideration, we feel that it has merit but does not fully meet PLOS ONE’s publication criteria as it currently stands. Therefore, we invite you to submit a revised version of the manuscript that addresses the points raised during the review process.

We look forward to receiving your revised manuscript.

Kind regards,

Stefano Piraino

Academic Editor

PLOS ONE

Journal Requirements:

“We are grateful to the Brazilian National Institute of Science and Technology for Tropical Marine Environments (INCT AmbTropic), the Brazilian Research Network on Global Climate Change (Rede CLIMA) and European Integrated CARBOCHANGE for funding the Camadas Finas III survey and to all the scientific team on board. We thank the CNPq (Brazilian National Council for Scientific and Technological Development), which provided a PhD scholarship to E.G.T. (grant 140897/2017-8) and a Research Scholarship to S.N.L. and M.A. This work is a contribution to the LMI TAPIOCA (www.tapioca.ird.fr), CAPES/COFECUB program (88881.142689/2017-01), the European Union’s Horizon 2020 projects PADDLE (grant agreement No. 73427) and TRIATLAS (grant agreement No. 817578).”

“The author received no specific funding for this work.”

5. We note that Figures 1, 2, 4, 5, 6, 7 and 9 in your submission contain map images which may be copyrighted. All PLOS content is published under the Creative Commons Attribution License (CC BY 4.0), which means that the manuscript, images, and Supporting Information files will be freely available online, and any third party is permitted to access, download, copy, distribute, and use these materials in any way, even commercially, with proper attribution. For these reasons, we cannot publish previously copyrighted maps or satellite images created using proprietary data, such as Google software (Google Maps, Street View, and Earth). For more information, see our copyright guidelines: http://journals.plos.org/plosone/s/licenses-and-copyright.

 a. You may seek permission from the original copyright holder of Figures 1, 2, 4, 5, 6, 7 and 9 to publish the content specifically under the CC BY 4.0 license. 

Reviewers' comments:

Reviewer's Responses to Questions

**Comments to the Author**

1. Is the manuscript technically sound, and do the data support the conclusions?

Reviewer #1: Yes

Reviewer #2: Yes

Reviewer #3: Partly

2. Has the statistical analysis been performed appropriately and rigorously? 

Reviewer #1: Yes

Reviewer #2: Yes

Reviewer #3: Yes

3. Have the authors made all data underlying the findings in their manuscript fully available?

Reviewer #1: Yes

Reviewer #2: Yes

Reviewer #3: Yes

4. Is the manuscript presented in an intelligible fashion and written in standard English?

Reviewer #1: Yes

Reviewer #2: Yes

Reviewer #3: No

5. Review Comments to the Author

Reviewer #1: This manuscript reports for the first time the effect of the Amazon River plume (ARP), one of the largest river discharge plumes in the world, on the planktonic cnidarian community composition. By sampling across a huge swathe of variation in oceanographic conditions, this study provides extremely valuable insights on the hydrographic factors that drive the distribution of many understudied gelatinous species.

PLos One is a general audience journal and therefore the authors should define specialized oceanographic jargon (such as retroflect) as well as avoid shortcuts with implicit units and assumptions on the audience's knowledge of how a plankton net works and what different meshes refer to, for example. The RDA shows confounding distribution of factors between the mesoscale features (i.e. cyclonic and anticyclonic eddies) and the distribution of oceanic/neritic zones with Neritic and ARP vectors. This may be because all the observed eddies occur exclusively in some areas. Perhaps a subsetted analysis with only neritic or ARP/non-ARP sample data could better resolve their influence on planktonic cnidarians.

Line by line:

27: we concluded -> we hypothesized

99: define retroflection

114: remove space before period .

118: achieved -> obtained

120: under a stereomicroscope; double "and"

122: provided -> published/reported

133: give units to 35 and 22

134: give units to 24.5

141: in samples from nets with ... meshes. Thus we merged the data from both meshes for statistical analysis.

142: unique -> only

158: avoid starting sentences with numbers i.e. Zooplankton biomass from nets with 120µm ...

159: availability for planktonic cnidarians

174: and stations

182: give units to 35

185: (mean of samples collected with 120 and 300 μm mesh nets)

189: units for 35

192: units for 36...

194: units for 35

209: florescence -> fluorescence

246: number of species -> species richness

247: first -> top

262: see comment for line 185

283: see comment for line 185

321: A Monte Carlo

383: metabolism -> metabolic demand

385: is turbid -> has low visibility

386: take vantage from other competitors -> outcompete visual predators

400, 402: M. kocchii -> M. kochii (perhaps also elsewhere in the manuscript)

432: schscholtzii -> esschscholtzii

435: units for 32.5

436: units for 36

459: conclude -> hypothesize (this study does not formally test the mechanistic hypothesis behind the biogeographic correlations)

463: kochi -> kochii

Fig. 2: SSS needs units

Figures all have poor image resolution, perhaps this is just for the review version.

Reviewer #2: I believe that the work makes an exceptional contribution to science, considering that this type of research and the impact of the Amazon River on the environment in this way has not been researched or published until now. It is necessary to make corrections of the English language and with minor changes I propose that this paper be published in the journal PLOS ONE.

Line 31-33 – I this system Persa incolorata…. were representative species.

Line 37- We concluded…

Line 49- Clarify sentences

Line 88-90- Reorganize this sentences

Line 158- exclude to start sentence with number

Line 184- In section “Environmental background –Figure 2 zooplankton biomass is mentioned. It is better to separate environmental parameters from biological parameters

Line 196-197- All collected data are from October 2012

Line 250-251 Reorganize sentence

Line 448-450 Clarify more…

Table 2: Typical default for statistical packages is to print three decimals (for p value)

Line 368-378 - Try to leave out the use of Figs in the discussion. If the description of the figures is in question, it has its place in the results. Move part of the discussion to the results, and then in the discussion focus on comparing the results and drawing conclusions.

Reviewer #3: Please see the attached file.

6. PLOS authors have the option to publish the peer review history of their article (what does this mean?). If published, this will include your full peer review and any attached files.

Reviewer #1: No

Reviewer #2: No

Reviewer #3: No

---

## [Author Response · Author response to Decision Letter 0]

30 Jun 2023

Reviewer comments

Reviewer #1: This manuscript reports for the first time the effect of the Amazon River plume (ARP), one of the largest river discharge plumes in the world, on the planktonic cnidarian community composition. By sampling across a huge swathe of variation in oceanographic conditions, this study provides extremely valuable insights on the hydrographic factors that drive the distribution of many understudied gelatinous species.

PLos One is a general audience journal and therefore the authors should define specialized oceanographic jargon (such as retroflect) as well as avoid shortcuts with implicit units and assumptions on the audience's knowledge of how a plankton net works and what different meshes refer to, for example

 The RDA shows confounding distribution of factors between the mesoscale features (i.e. cyclonic and anticyclonic eddies) and the distribution of oceanic/neritic zones with Neritic and ARP vectors. This may be because all the observed eddies occur exclusively in some areas. Perhaps a subsetted analysis with only neritic or ARP/non-ARP sample data could better resolve their influence on planktonic cnidarians.

R: Indeed, eddies occurred in specific habitats. Although the eddies vectors were not useful to explain species distribution, we decided to keep them since one of the objectives of such analyses is to show the environmental structure of co-occurring factors.

We tried to perform the analysis isolating each environment. However, the number of samples is relatively low for some of the environments hampering a more accurate association, and we do not believe it would bring significant improvements if included in the manuscript

Line by line:

27: we concluded -> we hypothesized

R: We changed it.

99: define retroflection

R: We included “(i.e. a change in the flow direction)” after it.

114: remove space before period .

R: We removed it.

118: achieved -> obtained

R: We changed it.

120: under a stereomicroscope; double "and"

R: The second and refers to specimens, so it is correct.

122: provided -> published/reported

R: We changed it.

133: give units to 35 and 22

R: Salinity, when measured from conductivity, has no unity. We included (σT) for density.

134: give units to 24.5

R: We included it.

141: in samples from nets with ... meshes. Thus we merged the data from both meshes for statistical analysis.

R: We corrected it

142: unique -> only

R: We changed it.

158: avoid starting sentences with numbers i.e. Zooplankton biomass from nets with 120µm ...

R: We changed it

159: availability for planktonic cnidarians

R: We included it.

174: and stations

R: We included it

182: give units to 35

R: See response for line 133.

185: (mean of samples collected with 120 and 300 μm mesh nets)

R: We changed it.

189: units for 35

R: See response for line 133.

192: units for 36...

R: See response for line 133.

194: units for 35

R: See response for line 133.

209: florescence -> fluorescence

R: We corrected it.

246: number of species -> species richness

R: We changed it.

247: first -> top

R: We changed it

262: see comment for line 185

R: We changed it.

283: see comment for line 185

R: We changed it.

321: A Monte Carlo

R: We changed to “The Monte Carlo”

383: metabolism -> metabolic demand

R: We changed it

385: is turbid -> has low visibility

R: We changed it.

386: take vantage from other competitors -> outcompete visual predators

R: We changed it

400, 402: M. kocchii -> M. kochii (perhaps also elsewhere in the manuscript)

R: We corrected it and verified the entire manuscript. 

432: schscholtzii -> esschscholtzii

R: We corrected it (eschscholtzii).

435: units for 32.5

R: See response for line 133.

436: units for 36

R: See response for line 133.

459: conclude -> hypothesize (this study does not formally test the mechanistic hypothesis behind the biogeographic correlations)

R: We changed it.

463: kochi -> kochii

R: We corected it.

Fig. 2: SSS needs units

R: See response for line 133.

Figures all have poor image resolution, perhaps this is just for the review version.

R: Yes, it is in the review version only. 

Reviewer #2: I believe that the work makes an exceptional contribution to science, considering that this type of research and the impact of the Amazon River on the environment in this way has not been researched or published until now. It is necessary to make corrections of the English language and with minor changes I propose that this paper be published in the journal PLOS ONE.

R: 

Line 31-33 – I this system Persa incolorata…. were representative species.

R: We included it.

Line 37- We concluded…

R: We changed it

Line 49- Clarify sentences

R: It seems clear for us. Any suggestion?

Line 88-90- Reorganize this sentences

R: We reorganized it.

Line 158- exclude to start sentence with number

R: We changed it.

Line 184- In section “Environmental background –Figure 2 zooplankton biomass is mentioned. It is better to separate environmental parameters from biological parameters

R: Thanks for the suggestion, but we prefer to keep as it is. Indeed, zooplankton biomass was included as a proxy of the food availability for cnidarians. Thus, it belongs to cnidarians environmental background, which is not exclusively physical.

Line 196-197- All collected data are from October 2012

R: We changed it.

Line 250-251 Reorganize sentence

R: We reorganized it.

Line 448-450 Clarify more…

R: We improved the text.

Table 2: Typical default for statistical packages is to print three decimals (for p value)

R: We changed the p-values to three decimals.

Line 368-378 - Try to leave out the use of Figs in the discussion. If the description of the figures is in question, it has its place in the results. Move part of the discussion to the results, and then in the discussion focus on comparing the results and drawing conclusions.

R: Thanks for your suggestion, we removed reports of results and figures from discussion and kept only the strictly necessary.

Reviewer #3: Please see the attached file.

Comments in the attached file:

Line 19: Abstract must be abbreviated. Only most important facts should be present. Conclusion is very general. A stronger conclusion is needed related to the plum and currents in the area.

R: Thanks for your suggestion. We reduced the abstract and improved the conclusion there.

Line 124: Siphonophores were rarely collected intact so it is very hard to count colonies of syphonophores, especially for Physonecte.

R: We agree in the case of physonects, that is why we used the number of calicophoran anterior nectophores and bracts, and physonect nectophores to estimate the number of colonies. See the next comments for a detailed discussion on the subject.

Line 126: The eudoxid bracts are small and could be easily lost. The better way is to count the free gonophores.

R. Thanks for the suggestion. We disagree that it would be better to count free gonophores. First, most of them are not described, so it would impossible to reach the taxonomic levels we achieved based on them. Also, it is not hard to find eudoxids with more than one gonophore, thus counting them could lead to erroneous abundances of colonies. Furthermore, among dominant Calicophora species, only C. appendiculata and M. kocchii have bracts considerably smaller than other structures. Some of them may have been lost during trawls, however mesh selectivity is a potential bias in every zooplankton sampling based on nets.

Published studies with same approach in relation to bracts: https://doi.org/10.1093/plankt/fby001;
https://doi.org/10.1017/S0025315416001302

 Also, what is with intact eudoxids? Why they were not counted? The whole eudoxids and gonophores must be enumerated for the eudoxid abundance.

R: Intact eudoxids (as well as polygastrics) were counted as one colony, as in the case of free bracts.

It would be most accurate to show the number of nectophores per cubic meter as well as the number of gonophores (eudoxia) per cubic meter.

R: Sorry, but we do not see your point here. One nectophore (anterior) is equal to one polygastric colony and one bract (again, bracts are better than gonophores) is equal to one eudoxid colony.

Line 128: I suggest put only the qualitative analyses of physonectes, not quantitative. Dividing by 10 is not reliable for quantitative analysis and conclusions in your paper, particulary when you compared their abundance to other cnidarians.

R: Thanks for the suggestion, although we agree that dividing the number of physonect pneumatophore by 10 is a gross estimate it is a general rule in most recent planktonic cnidarians studies.

e.g.: https://doi.org/10.1371/journal.pone.0196431;
https://doi.org/10.1093/plankt/fby001

Additionally, only N. bijuga, A. elegans and A. okeni were included in multivariate analyses. And the three were quite less abundant than dominant species, thus removing then would not change our results and conclusions, while including them may provide some insights on their habitat preferences.

Line 362: Too long, shoul be shortened at least 30%.

Remove the reference to the figures from the discussion as well as the figure 9.

Do not repeat the results. The discussion should be concise, clear and with stronger conclusions.

Check the typefeller.

R: Thanks for your suggestion, we removed reports of results and figures from discussion and kept only the strictly necessary. However we prefer to keep Figure 9.

Line 454: If this part is conclusion you have to put title above. In any case, the sentences and conclusions already stated above in the discussion are repeated here. Nowhere were species analyzed as possible indicators of specific zones. If this paragraph is a conclusion, then it should be written more concisely and clearly, without repetition and unnecessary re-listing of species.

R: Thanks for your suggestion. We removed repeated discussions and made the conclusion more concise.

Table 1:

Separate Calycophorae and Physonecte

R: Since taxonomy is not the focus of this study, we do not see the relevance of a more systematics in the table. In any case the complete classification of all species from this study are available in Tosetto et al. (2022).

Tosetto EG, Neumann-Leitão S, Araujo M, Nogueira Júnior M. Epipelagic cnidarian fauna in the Western Equatorial Atlantic Ocean off the Amazon River Delta. Mar Biodivers. 2022;52: 50. doi:10.1007/s12526-022-01286-0

Taxa names not correct: 

R: These are specimens that were not identified up to genus level, in such cases, both “gen. sp.” and “sp.” are accepted.

---

## [Decision Letter · Decision Letter 1]

13 Aug 2023

Amazon River plume habitats shape planktonic cnidarian assemblages in the Western Atlantic

PONE-D-23-10398R1

Dear Dr. Tosetto,

We’re pleased to inform you that your manuscript has been judged scientifically suitable for publication and will be formally accepted for publication once it meets all outstanding technical requirements.

Kind regards,

Vitor Hugo Rodrigues Paiva, Ph.D.

Academic Editor

PLOS ONE

Additional Editor Comments (optional):

Reviewers' comments:

Reviewer's Responses to Questions

**Comments to the Author**

1. If the authors have adequately addressed your comments raised in a previous round of review and you feel that this manuscript is now acceptable for publication, you may indicate that here to bypass the “Comments to the Author” section, enter your conflict of interest statement in the “Confidential to Editor” section, and submit your "Accept" recommendation.

Reviewer #1: All comments have been addressed

Reviewer #2: All comments have been addressed

2. Is the manuscript technically sound, and do the data support the conclusions?

Reviewer #1: Yes

Reviewer #2: Yes

3. Has the statistical analysis been performed appropriately and rigorously? 

Reviewer #1: Yes

Reviewer #2: Yes

4. Have the authors made all data underlying the findings in their manuscript fully available?

Reviewer #1: Yes

Reviewer #2: Yes

5. Is the manuscript presented in an intelligible fashion and written in standard English?

Reviewer #1: Yes

Reviewer #2: Yes

6. Review Comments to the Author

Reviewer #1: All my comments have been adequately addressed and minor issues have been corrected. I can now recommend this manuscript for publication.

Reviewer #2: (No Response)

7. PLOS authors have the option to publish the peer review history of their article (what does this mean?). If published, this will include your full peer review and any attached files.

Reviewer #1: No

Reviewer #2: No

---

## [Editor Report · Acceptance letter]

17 Aug 2023

PONE-D-23-10398R1 

Amazon River plume habitats shape planktonic cnidarian assemblages in the Western Atlantic 

Dear Dr. Tosetto:

I'm pleased to inform you that your manuscript has been deemed suitable for publication in PLOS ONE. Congratulations! Your manuscript is now with our production department. 

Kind regards, 

on behalf of

Dr. Vitor Hugo Rodrigues Paiva 

Academic Editor

PLOS ONE